# Single-Walled Carbon Nanotube Synthesis Yield Variation in a Horizontal Chemical Vapor Deposition Reactor

**DOI:** 10.3390/nano11123293

**Published:** 2021-12-04

**Authors:** Sung-Il Jo, Goo-Hwan Jeong

**Affiliations:** Department of Advanced Materials Science and Engineering, Kangwon National University, Chuncheon 24341, Gangwon-do, Korea; sungil107@kangwon.ac.kr

**Keywords:** single-walled carbon nanotubes, synthesis yield, horizontal CVD, methane, Raman spectroscopy

## Abstract

The controlled synthesis of single-walled carbon nanotubes (SWNTs) is essential for their industrial application. This study investigates the synthesis yield of SWNTs, which depends on the positions of the samples on a horizontal chemical vapor deposition (CVD) system. Methane and Fe thin films were used as the feedstock and catalyst for SWNTs synthesis, respectively. A high-resolution scanning electron microscope was used to examine the synthesis yield variation of the SWNTs along the axial distance of the reactor. The morphology and crystallinity of the fabricated SWNTs were evaluated by atomic force microscopy and Raman spectroscopy, respectively. We observed that the highest synthesis yield of the SWNTs was obtained in the rear region of the horizontal reactor, and not the central region. These results can be applied to the synthesis of various low-dimensional nanomaterials, such as semiconducting nanowires and transition metal dichalcogenides, especially when a horizontal CVD chamber is used.

## 1. Introduction

Single-walled carbon nanotubes (SWNTs) have a wide range of applications in various fields because of their outstanding physical, chemical, and electrical properties [1]. From an industrial point of view, the controlled synthesis of SWNTs is crucial in terms of structures and yield; therefore, a significant amount of work is dedicated to this topic. For instance, the specific growth of semiconducting or metallic SWNTs is highly desirable for device applications such as interconnections, electrodes, sensors, and other functional elements where superior electrical properties are required [2,3,4,5,6,7,8,9,10]. However, the growth of horizontally aligned (HA) SWNTs with specific chiralities is a representative issue in terms of achieving the above-mentioned goal [11].

Among the various methodologies used for SWNT growth, chemical vapor deposition (CVD) is widely employed because of its facile operation and ease of process control. Moreover, the vapor-liquid-solid (VLS) mechanism can be adopted to explain the procedure of SWNT synthesis in CVD. Semi-liquid catalytic metal particles preferentially incorporate carbon species from thermally dissociated feedstock and precipitate as hexagonal networks of supersaturated carbons that yield carbon nanotubes [12]. To date, the majority of syntheses using conventional horizontal CVD systems were performed through the installation of growth substrates at the chamber center, which provides a maximum temperature and high thermal stability.

Recently, we observed unexpected variations in the growth yields of SWNTs along the CVD reactor and performed a systematic investigation of these yield variations. To evaluate the obtained results, we investigated various synthesis parameters, such as gas flow rate and growth temperature. In fact, the variation of growth features of vertically aligned multi-walled carbon nanotubes (VA-MWNTs) was investigated by some groups [13,14,15,16,17]. Because the growth behavior of SWNTs is significantly influenced by the minute change in growth environment such as gas flow rate and growth temperature, we focused on the growth of SWNTs with respect to the sample position. This study could be expanded to the understanding of the density-controlled growth of other functional nanomaterials, such as ZnO nanowires for sensor devices and Si nanowires for energy harvesting applications.

## 2. Materials and Methods

### 2.1. CVD Chamber

Figure 1a,b show the schematic illustration and digital photo of the employed CVD chamber, respectively. The gases used for SWNT growth were introduced into a horizontal quartz tube. The diameter of the tube and the length of the heating zone were 2.5 and 20 cm, respectively [18,19]. To investigate the growth behavior of the SWNTs with respect to their position on the CVD chamber, growth substrates were installed throughout the furnace zone, as indicated in red in Figure 1b. To analyze the effect of chamber size on the growth behavior, we employed a different CVD quartz furnace with a 5 cm diameter and 40 cm-long heating zone.

### 2.2. Substrate Preparation and SWNT Growth on SiO_2_ and Quartz Substrates

We prepared Fe thin films of 1-Å thickness on silicon oxide wafers and ST-cut quartz wafers (MTI-Korea) as growth substrates for SWNTs. ST-cut quartz, a type of Y-cut quartz with a 38° cut angle, was previously used to grow horizontally aligned SWNTs using strong interactions between carbon nanotubes and quartz substrates [19,20,21]. Based on a procedure described in a previous study, we first cleaned the as-purchased ST-cut wafers in acetone and thermally annealed them at 900 °C for 8 h in air to produce clear step structures on the quartz surface [19,20,21]. The substrates were carefully covered with Fe thin films using the electron-beam evaporation method at a deposition rate of 0.1 Å/s. The substrates were then inserted in the chamber. 

The Fe-deposited substrates were then oxidized at 900 °C in air for 10 min and cooled to below 200 °C. The chamber was successively evacuated twice to 5 × 10^−2^ Torr under Ar flow, and heated to 900 °C under Ar (225 sccm) and H_2_ (25 sccm) mixtures. After stabilizing the chamber at 900 °C for 10 min, a mixture of CH_4_ and H_2_ was introduced to grow the SWNTs for 10 min. After the growth process, the CVD furnace was cooled to ambient temperature under Ar flow. To investigate the effect of the total gas flow rate and growth temperature at each sample position inside the CVD chamber, the gas flow rate was varied from 150 to 500 sccm, while maintaining a mixing ratio of H_2_:CH_4_ = 10:90%. The growth temperature was varied from 850 to 925 °C. 

### 2.3. Characterization

A high-resolution scanning electron microscope (HRSEM, Hitachi S-4800) was used to analyze the SWNT growth with respect to sample positions within the CVD chamber. To quantitatively compare the yields of the SWNTs, we estimated their areal densities from the image-treated HRSEM images as follows:(1)$$Areal\;density\;of\;SWNTs\;\left(\%\right)=\frac{Area\;covered\;by\;SWNTs}{Total\;area\;of\;SEM\;image}\times 100\;$$

All of the original HRSEM images were modified using the B/W invert function in the PicMan software (Wafer Masters Inc., Dublin, CA, USA). When auto contrast was applied to the SEM image, 256 levels of brightness values were generated based on the highest and lowest brightness values within the total area. We adjusted the threshold value of the B/W channel so that the areas of the images that contained SWNTs were clearly distinguished. As a result, the image was composed of black and white pixels with brightness values of 0 and 255, respectively. Only the synthesized SWNT area on the image was white. The areal density was then calculated by comparing the number of white pixels covered by SWNTs with the number of pixels in the total area of the SEM image. Detailed examples of low (12–18%), medium (48–50%), and high (93–94%) synthesis yields are shown in the Appendix A.

Raman spectroscopy (LabRAM Aramis, Horiba, Tokyo, Japan) is a convenient tool for quickly estimating the diameter of SWNTs using the specific relation between the tube diameter and peak position in the radial breathing mode (RBM) region. It is also useful for determining the structural integrity of the samples by comparing the peak intensities of the structural-disorder-induced peak around 1350 cm^−1^ (D-band, I_D_) and the tangential stretching vibration mode of graphite around 1590 cm^−1^ (G-band, I_G_) [22,23,24]. The excitation wavelength of the Raman laser was 532 nm, and the spot size was 1 μm. The accuracy of the Raman frequency shift was calibrated within 1 cm^−1^ using the Si peak at 520 cm^−1^. 

Atomic force microscopy (AFM, Park Systems XE-70, Suwon, Korea) was used in the tapping mode to estimate the diameter and number of SWNTs per 1 μm, which were grown on the ST-cut quartz substrates. To investigate the possible application of the horizontally aligned SWNTs as transparent conductive electrodes, their optical transmittance was measured using UV-visible spectroscopy (Biochrom, Libra S80, Cambridge, UK) after their growth on the transparent quartz substrates.

## 3. Results and Discussion

### 3.1. Effects of Total Gas Flow Rate and Sample Position on SWNT Growth Yield

To investigate the effects of total gas flow rate and sample position on the SWNT growth yield, we performed a detailed HRSEM analysis after the growth of SWNTs on the SiO_2_ substrates. Based on a previous study [19], the gas composition and growth temperature were fixed at H_2_:CH_4_ = 10:90 and 900 °C, respectively. In the present study, the total amount of working gases changed from 150 to 500 sccm. Figure 2 shows the HRSEM results for different sample positions. The HRSEM images from the central region of the chamber are marked with a blue box. The densest SWNTs grown at each flow rate are indicated by red boxes. Notably, the substrate installed at the rear region contained a higher amount of SWNTs than the central region, under all gas flow rate conditions. In Figure 2, the densest SWNTs were observed at 10 cm behind the center using H_2_:CH_4_ = 25:225 sccm.

In general, the growth temperature is an essential parameter for determining the growth of the SWNTs. This is because each stage involving SWNT growth, including feedstock decomposition, catalyst activation, dissolution of carbon species into the catalyst, and precipitation of the supersaturated carbon into the hexagonal ring structure, depends on the thermal environment. Thus, we carefully measured the temperature profile of the inner space of the CVD chamber at 1 cm intervals using a K-type thermocouple after calibration. The temperature was measured after 5 min, during which it was held at each point to stabilize the environment. During measurement, the feedstock was changed to Ar. However, the total amount of Ar was kept equal to that used in the growth condition. According to the measurement results (Appendix A), the temperatures at the center and rear edge regions (10 cm behind the center) were 918 and 771 °C, respectively, when we set the chamber temperature to 900 °C. In addition, the temperature profiles were almost the same, regardless of the amount of Ar used, with perfect symmetry along the chamber axis. The central region showed the highest temperature of 918 °C and a relatively lower areal density of 86.12%. On the other hand, the rear edge region showed a relatively lower temperature of 771 °C but produced a higher growth density of 98.83% (Appendix A). 

These results suggest that the actual thermal environment at the rear end of the chamber may have been higher than the chamber center because of a preheated gas flow from the central region, which is introduced with the appropriate gas speed. As a result, the enhanced decomposition of feedstock gas due to the high thermal environment in that region results in a higher SWNT density. Similar results could be found from the vertical growth cases of MWNTs. Furthermore, an optimal amount of feedstock gas should be present to yield the highest SWNT density. If the carbonaceous species in the feedstock are depleted due to the high gas speed or lower gas amount, there is a decrease in the SWNT density or a shift in the highest growth region near the chamber center. In this study, when the total amount of feedstock was decreased to 200 and 150 sccm, a shift of the densest SWNT region was observed from 10 to 7.5 cm closer to the center. This result agrees with the above speculation. In the case of the H_2_:CH_4_ = 50:450 sccm condition, the density of SWNTs drastically decreased over the chamber. In addition, short and crumbled morphologies were observed, which were a general result of the over-injection of feedstock gases, resulting in the poisoning phenomena of catalytic particles [25].

We estimated the areal density of SWNTs through the aforementioned image treatment process using the obtained HRSEM images, and the quantitative results are presented in Figure 3. As confirmed in Figure 2, the highest areal density at all gas conditions was obtained at the rear end. rather than at the central region. of the CVD chamber. This is an unexpected result because conventional CVD is performed by loading the substrate at the center of the chamber. Here, the highest areal density of 99.25% was obtained for the sample installed 8.75 cm behind the center, using the H_2_:CH_4_ = 25 of 225 sccm gas condition. Moreover, we confirmed that lower flow-rate conditions, such as 200 and 150 sccm, led to the highest SWNT density region near the center. In addition, the poor growth of SWNTs under high flow-rate conditions (H_2_:CH_4_ = 50:450 sccm) was quantitatively verified, as shown in Figure 2.

The total amount of gases introduced into the chamber was a critical factor. However, the partial pressure of the carbon-containing gas remained constant because the feedstock determined not only the amount of carbon supply but also the residence time and flow type of the gases in the chamber. Here, we focused on the probable effects of the residence time and flow type of the gases because these may change the growth time and thermal environment in the chamber due to fast flow or turbulence of the gases. Thus, we estimated both the Reynolds and Knudsen numbers in our growth condition and compared these values to previously published results. 

The Reynolds number (*Re*) is defined as follows [26]:(2)$$Re=\frac{\rho vd}{\gamma }\;$$
where *ρ* is the gas density (kg/m^3^), *v* is the gas velocity (m/s), *d* is the diameter of the chamber (m), and *γ* is the viscosity of the gas (kgm^−1^ s^−1^).

The Knudsen number (*Kn*) is defined as follows [26]: (3)$$Kn=\frac{\lambda }{d}\;$$
where *λ* is the mean free path (m) and *d* is the diameter of the chamber (m).

As summarized in the Appendix A, the Reynolds and Knudsen numbers in this study range from 0.8 to 2.8 and 2.3 × 10^−34^ to 2.4 × 10^−34^, respectively. These values are in the range of laminar flow and are similar to the majority of previous results. In addition, we estimated the gas velocity by assuming the presence of an empty column in the CVD chamber as follows:(4)$$v=\frac{Volumetric\;flow\;rate}{\pi r^{2}}\;$$
where *volumetric flow rate* is expressed as mL/h and *r* is the radius of the chamber (m). 

In this study, we obtained the velocities of 0.66 to 2.19 cm/s. The values obtained in previous reports range from 0.7 to 3.51 cm/s, as shown in Appendix A. Based on the aforementioned comparison of gas-flow behavior, our growth conditions are very similar to those obtained in previous studies. 

### 3.2. Effects of Growth Temperature and the Sample Position on SWNT Growth Yield

As mentioned earlier, growth temperature is a key factor in determining growth behavior. Thus, we investigated the effect of growth temperature with a fixed feedstock composition of H_2_:CH_4_ = 25:225 sccm, based on the results of Section 3.1.

Figure 4 displays HRSEM images showing the growth results for SWNTs, depending on growth temperature and sample position. The growth temperature was varied from 850 to 925 °C. The HRSEM images at the center and highest growth density region are highlighted with blue and red boxes, respectively. We confirmed that the highest density was obtained at the rear part rather than the center of the CVD chamber at all growth temperatures. The densest SWNTs were also obtained 10 cm behind the center at 900 and 925 °C. Moreover, as the growth temperature decreased to 875 and 850 °C, the densest SWNTs were observed 7.5 and 5 cm behind the center, respectively.

We estimated the areal density of SWNTs based on the HRSEM images, as shown in Figure 5. Here, the highest areal density of 99.72% was obtained from the sample installed at the furnace edge (10 cm behind the center) at 925 °C. We also observed that the highest SWNT density region shifted to near the center of the chamber with decreasing growth temperatures. In addition, the areal density of the SWNTs was drastically diminished from the furnace edge (10 cm behind the center) at all growth temperatures. At this point, we may also explain the obtained results from the point of view of the actual thermal environment in the CVD chamber, which directly affects feedstock decomposition, catalyst activation, and the final growth of SWNTs. We checked the actual temperature variation at all growth temperatures using the same thermocouple (Appendix A). The temperature profiles also showed a high symmetry. The center region shows the highest temperature, and the temperature decreases at the furnace edge. The high density of SWNTs grown at 925 °C is indicated at the furnace edge region (10 cm behind the center), as shown in Appendix A.

We expected the rear part of the chamber to exhibit a higher temperature than that measured, owing to the gas flow. The inlet gases were heated from the front part of the chamber, and the temperature gradually increased as the gases passed through the highest temperature region, the chamber center, until the gases exited the furnace region. Thus, we obtained the highest SWNT density in the rear region as the growth temperature increased due to the enhanced thermal environment, in which active growth might have occurred. In addition, the expansion of the high-growth region with an areal density beyond 50% proves the presence of an elevated thermal environment through an increase in the growth temperature. At the temperatures of 800 and 850 °C, the sharp decrease in areal density at the furnace edge region may be attributed to the low temperature, which was below 750 °C, at the edge region, which may be too low to activate the growth process in the methane feedstock.

In summary, we determined that the maximum growth yield may be obtained in the rear region rather than at the center of the CVD chamber. In addition, the maximum growth yield point shifted to the furnace end region with increasing flow rate and growth temperature. An expansion of the range, yielding high-density SWNT growth at higher growth temperatures, is direct evidence of the enhanced thermal environment in the chamber. Furthermore, we confirmed the exact coincidence of the obtained results using a 2-inch-diameter CVD chamber, as shown in Appendix A.

### 3.3. SWNT Analysis Using Raman Spectroscopy

Raman spectroscopic analysis was performed to examine the structure and quality of the SWNTs. Figure 6 shows the representative Raman spectra for SWNTs grown at the center (Figure 6a) and the highest-density region (Figure 6b) at 900 °C with H_2_:CH_4_ = 25:225 sccm. We obtained the typical Raman spectra of SWNTs with RBM peaks and the tangential stretching mode from the graphite structure (G-band) around 1590 cm^−1^. It is difficult to observe the structural-disorder-related vibration mode (D-band) at approximately 1350 cm^−1^ for all the recorded spectra. In general, it is possible to determine the crystal integrity of SWNTs using the ratio of peak intensity (I_D_/I_G_) [22,23,24]. Thus, we estimated the average I_D_/I_G_ values from one hundred Raman spectra in the central and maximum growth density regions and found the ratios of peak intensity to be 0.017 and 0.022, indicating the extremely high crystalline integrity of the SWNTs. In fact, these values are significantly lower than those of high-quality commercial SWNTs (0.111). 

For the tube structure, we estimated the SWNT diameters from RBM peaks according to the relation (5) [22,23,24]:*ω* = 248/*d*(5)

Here, *ω* is the Raman frequency (cm^−1^) and *d* is the tube diameter (nm). Thus, from the obtained RBM peak positions, we found that the range of the SWNT diameter was 1.39–1.98 nm and 1.30~2.25 nm for the center and highest-tube-density regions, respectively. A relatively wide range of tube diameters in the highest-density region is generally acceptable because a higher thermal environment gives rise to a wider size distribution of the catalytic nanoparticles and corresponding tubes.

### 3.4. High-Density Growth of Horizontally Aligned SWNTs on Quartz Substrates

In this section, we demonstrate the dependence of SWNT density on sample position using an ST-cut quartz substrate that yielded HA-SWNTs. The HA-SWNT growth process on a quartz substrate is reported in the literature [19,21,27,28,29].

Figure 7 shows SEM and AFM results for a quartz wafer installed at the chamber center using the optimized growth condition, H_2_:CH_4_ = 25:225 sccm at 900 °C. The Fe catalytic thin films were coated on plain (a and c) and line-patterned (b and d) quartz wafers for comparison. For a quantitative comparison of the tube density, we defined the linear density of SWNTs as the number of tubes passing thorough a 1-μm length. We found that HA-SWNTs grew on all substrates, and that the linear density of the tube was approximately 7.3 and 7.7 tubes/μm for plain and line-patterned wafers, respectively. Figure 7e shows the AFM topographic data and corresponding height profile from the sample shown in Figure 7c. The majority of the tubes were within the diameter range of 1–2 nm, which agrees well with the Raman spectra for the SWNTs grown on SiO_2_ substrates. Heights of over 3 nm indicate the presence of catalytic nanoparticles on the wafer. 

Figure 8 displays the SEM and AFM results of the HA-SWNTs grown on the quartz wafer installed in the maximum-synthesis-yield region using the same growth conditions used for the sample shown in Figure 7. The HA-SWNTs were uniformly grown on the substrates, and the linear tube density was approximately 11.1 and 13.9 tubes/μm for plain and line-patterned wafers, respectively. Figure 8e shows an AFM image and the corresponding height profile for the sample shown in Figure 8c. RBM peaks in the Raman spectra for HA-SWNTs exist within 140–170 cm^−1^, which correspond to a diameter of 1.46–1.77 nm according to the above-mentioned relation (Appendix A). Considering that the diameter distribution of SWNTs on SiO_2_ substrates had a relatively wide range of 1.30–2.25 nm, the quartz wafer yielded a relatively narrow diameter distribution compared to the SiO_2_ substrates. Because Raman analysis presents limitations in evaluating the diameter distribution of SWNTs due to the specific resonance characteristic existing between the excitation wavelength of the laser and tubes, further analysis using lasers with different wavelengths is necessary for detailed characterization. In addition, the synthesized HA-SWNTs are found to be metallic tubes according to the Kataura plot [30]. 

Because SWNTs grown on quartz are highly aligned with a metallic character, they can be applied as transparent conductive films (TCF). Figure 9a shows digital photographs of quartz substrates taken before and after SWNT growth under the optimized conditions. However, it is difficult to determine differences in brightness with the naked eye. Thus, we measured the transmittance of the samples after SWNT growth using UV-vis spectroscopy. After normalizing the transmittance using a bare quartz substrate, we quantitatively obtained the transmittance of visible rays at 550 nm, as shown in Figure 9b. As a result, we found that the transmittances of the quartz from the center and maximum growth yield regions were 97.8 and 96.4%, respectively. As a further work, we are investigating the trade-off relation between transmittance and planar conductivity, in order to fabricate the relevant TCF devices.

Finally, to achieve the plausible application of SWNTs in sensors, energy harvesting or storage technologies, and electronics, the present study should be extended to various low-dimensional functional nanomaterials, such as one-dimensional semiconducting nanowires (ZnO, Si, SnO, BN, etc.) and two-dimensional dichalcogenides (MoS_2_, MoSe_2_, WS_2_, WSe_2_, etc.). In addition, detailed theoretical simulations from fluid dynamics studies are required. However, we still believe that the current results on HA-SWNTs suggest their potential use in future applications, such as transmission conducting films or sensor device fabrication. 

## 4. Conclusions

A detailed investigation of the synthesis yield variations of SWNTs was conducted with respect to their sample position in a horizontal CVD chamber. Methane feedstock and Fe catalytic thin films were used to synthesize the SWNTs. Based on the results of SEM, AFM, and Raman spectroscopy, we observed that the highest synthesis yield of SWNTs was obtained in the rear region of the horizontal reactor, and not in the central region. Identical results were obtained when we employed quartz substrates and different CVD chambers with larger diameters. A higher SWNT density was obtained in the rear region rather than the center due to the presence of an enhanced thermal environment in the chamber, which facilitated the decomposition of feedstock with a gas flow. The obtained results may be applied to the synthesis of various functional nanomaterials, such as semiconducting nanowires and transition metal dichalcogenides, especially in cases where a horizontal CVD chamber is employed.

## Figures and Tables

**Figure 1 nanomaterials-11-03293-f001:**
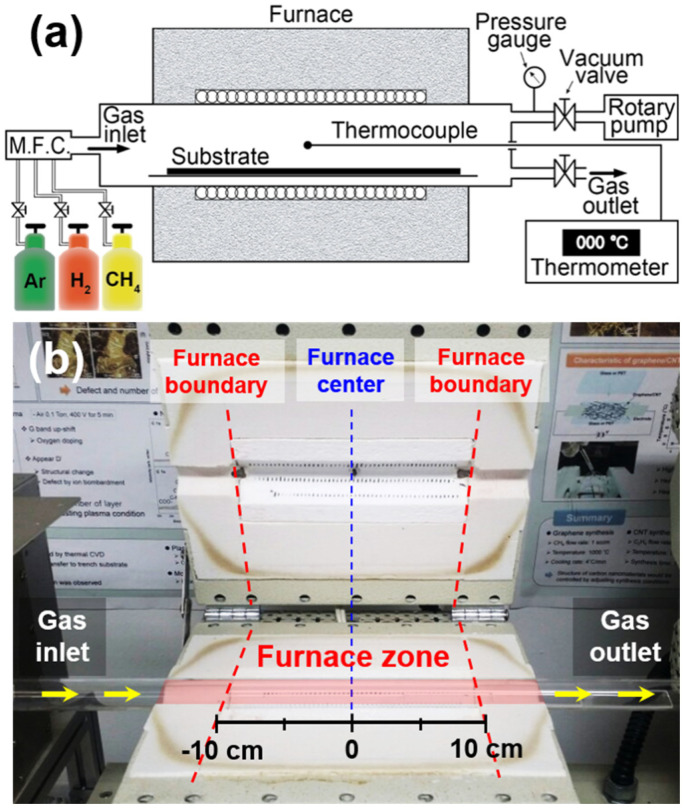
(**a**) Schematic illustration of the CVD chamber. (**b**) Digital photo of the apparatus.

**Figure 2 nanomaterials-11-03293-f002:**
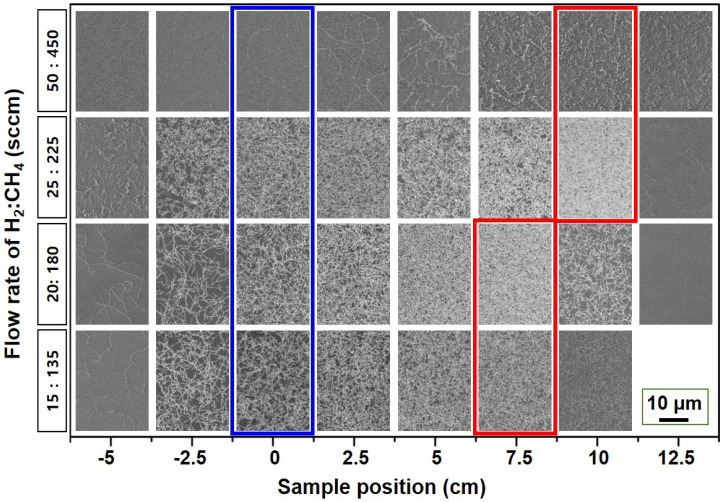
HRSEM images showing the growth results for SWNTs, depending on gas flow rate and sample position. SEM images from the central region and maximum-synthesis-yield regions are highlighted with blue and red boxes, respectively.

**Figure 3 nanomaterials-11-03293-f003:**
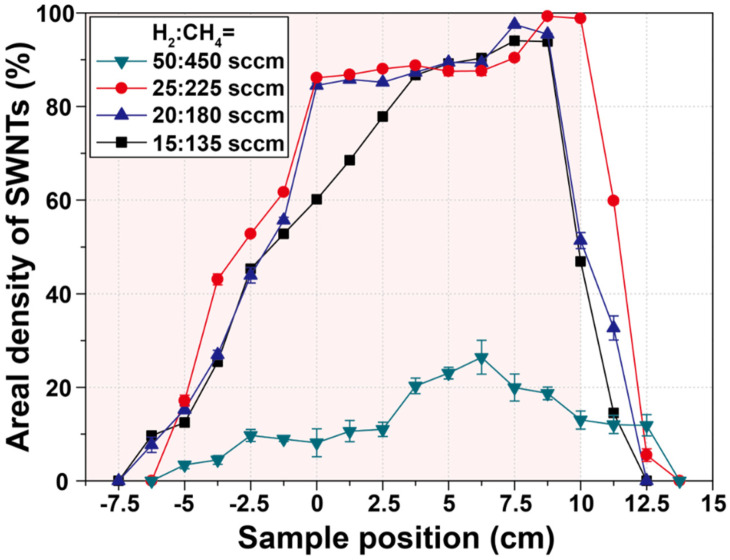
Variation of the areal density of SWNTs with respect to gas flow rate and sample position.

**Figure 4 nanomaterials-11-03293-f004:**
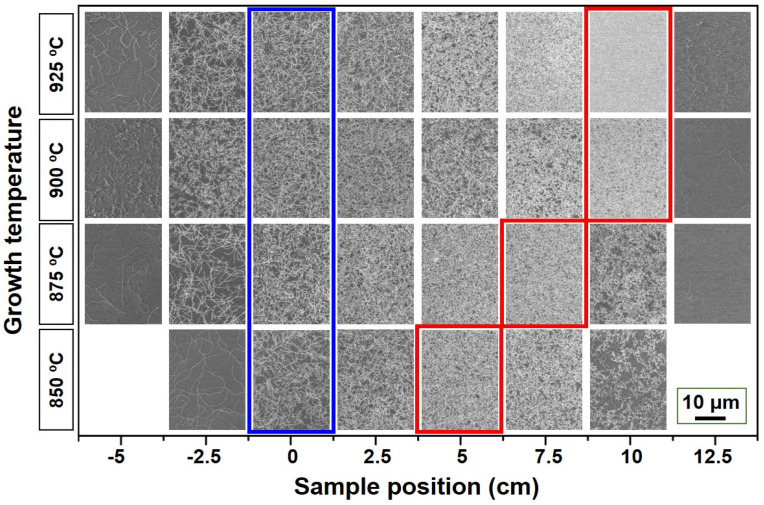
HRSEM images showing the growth results for SWNTs depending on the growth temperature and sample position. SEM images from the central region and maximum-synthesis-yield regions are highlighted with blue and red boxes, respectively.

**Figure 5 nanomaterials-11-03293-f005:**
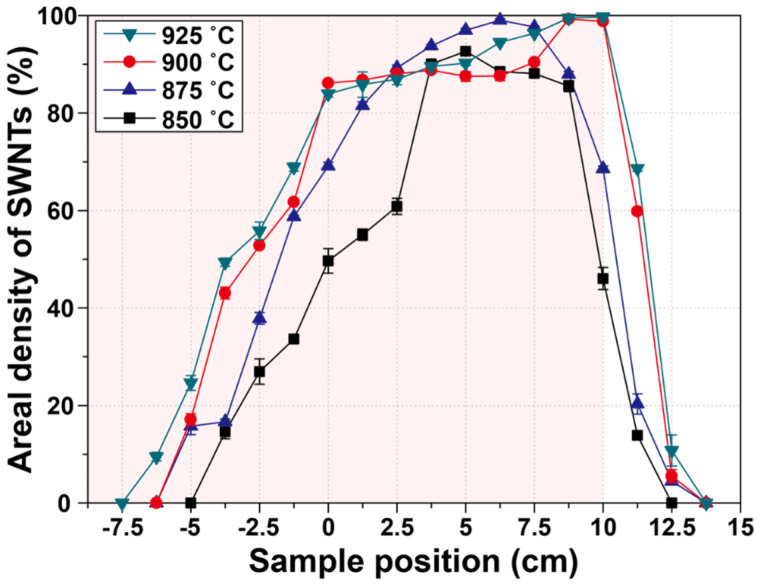
Variation of the areal density of SWNTs with respect to growth temperature and sample position.

**Figure 6 nanomaterials-11-03293-f006:**
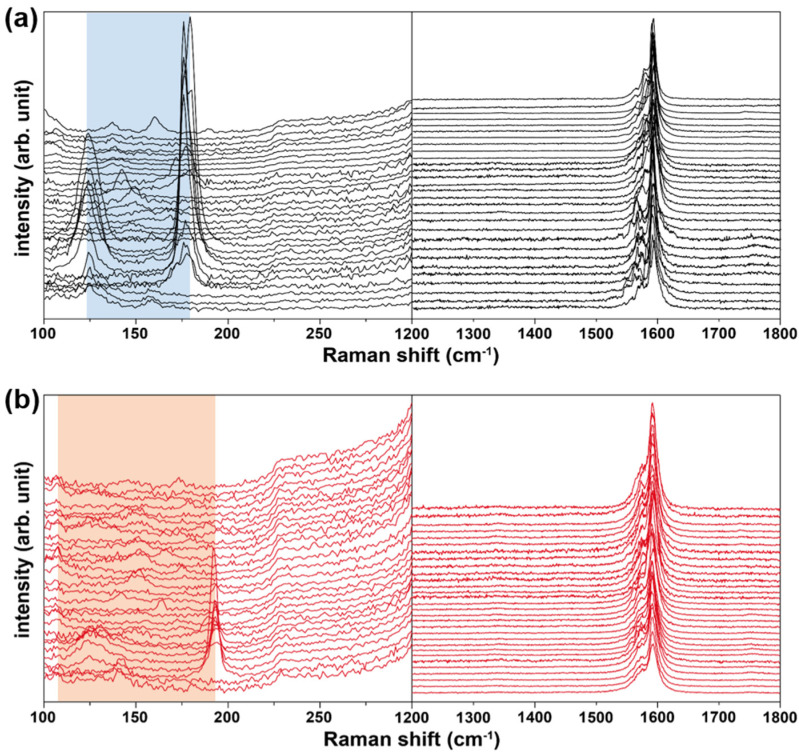
Raman spectra recorded for the (**a**) central region and (**b**) maximum-synthesis-yield regions. The sample was grown at 900 °C with H_2_:CH_4_ = 25:225 sccm. Regions where RBM peaks were detected are denoted by blocks of color.

**Figure 7 nanomaterials-11-03293-f007:**
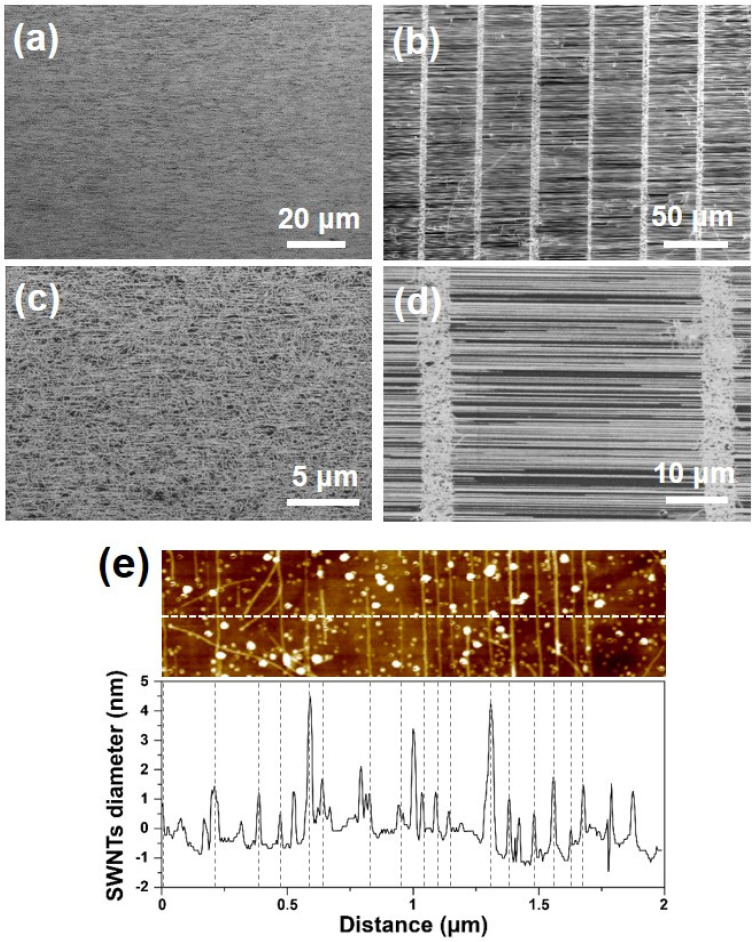
SEM images of SWNTs taken at the central region of plain (**a**,**c**) and line-patterned (**b**,**d**) ST-cut quartz wafers. (**e**) AFM topographic image and corresponding height profile result from (**d**). The sample was grown at 900 °C with H_2_:CH_4_ = 25:225 sccm.

**Figure 8 nanomaterials-11-03293-f008:**
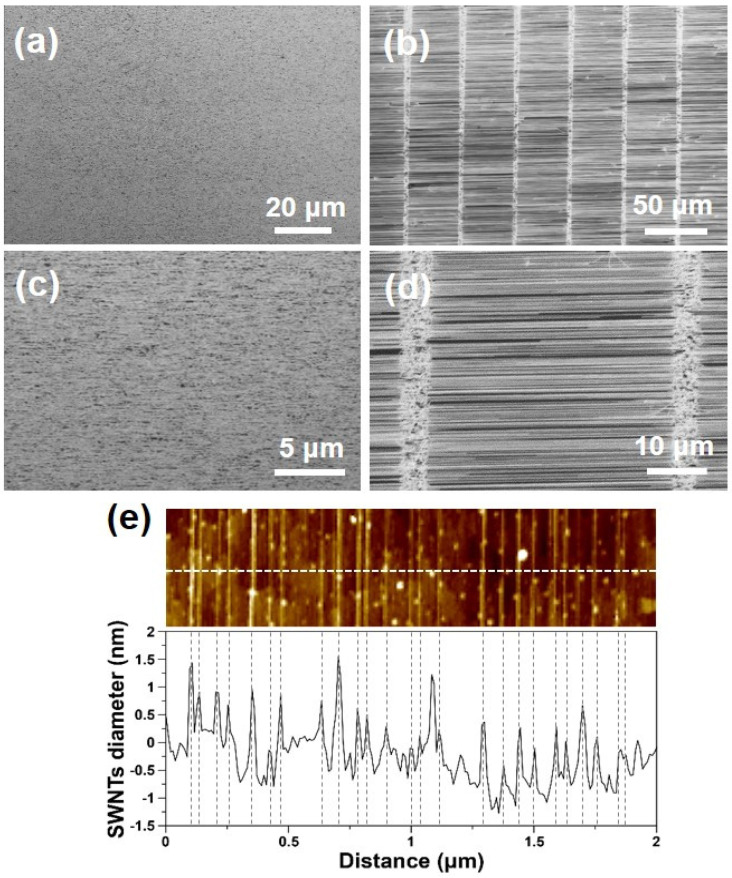
SEM images of SWNTs taken at the maximum-synthesis-yield region of plain (**a**,**c**) and line-patterned (**b**,**d**) ST-cut quartz wafers. (**e**) AFM topographic image and corresponding height profile obtained from (**d**). The sample was grown at 900 °C with H_2_:CH_4_ = 25:225 sccm.

**Figure 9 nanomaterials-11-03293-f009:**
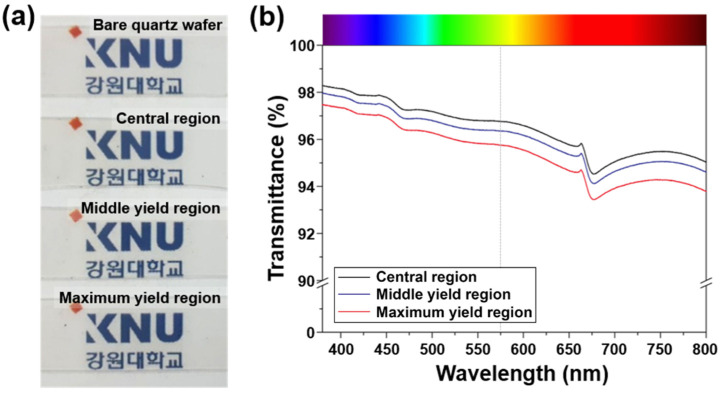
(**a**) Digital photo showing the different transmittances of SWNTs on quartz wafers. (**b**) UV-vis spectra recorded for a ST-cut quartz wafer in different synthesis yield regions.

## Data Availability

Not applicable.

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
