# Peer review of "Single-Walled Carbon Nanotube Synthesis Yield Variation in a Horizontal Chemical Vapor Deposition Reactor"

_nanomaterials, 2021, doi:10.3390/nano11123293_

Round 1

Reviewer 1 Report

The results are of potential interests to the readers in field of nanomaterials. The manuscript is recommended for publication after addressing the following issues.

(a) The manuscript should be thoroughly revised in English. The authors are suggested to invite a native speaker to polish the manuscript. 

(b) All equations/formulas should be numbered.

(c) Labels in some figures should be enlarged (e.g., Fig. 1, Fig. 2, ......).

Author Response

The results are of potential interests to the readers in field of nanomaterials. The manuscript is recommended for publication after addressing the following issues.

(a) The manuscript should be thoroughly revised in English. The authors are suggested to invite a native speaker to polish the manuscript. 
--> The manuscript is revised by native speaker as suggested by the Reviewer 1.

(b) All equations/formulas should be numbered.
--> We thank for the comment and revised.

(c) Labels in some figures should be enlarged (e.g., Fig. 1, Fig. 2, ......).
--> Some (Fig 1, Fig 3, Fig 5, and Fig 9) are replaced by new figures with larger font

Reviewer 2 Report

Taken into account all the following shortcomings mentioned, this paper needs to be revised.

  1. Language expression. Please polish the language and check for grammatical errors to make the sentence more readable and logical.
  2. In Figures 2 and 4, why didn't you add SEM images of the -12.5~-5 cm location?
  3. The temperature inhomogeneity at the horizontal position of the CVD furnace is not a favorable factor for the growth of single-walled carbon nanotubes, especially for the batch growth of single-walled carbon nanotubes. Can a reasonable method be proposed to optimize the temperature distribution? What is the innovation of this article?
  4. In lines 138 and 139, the expression of the sentence does not seem to be very accurate. The middle of the CVD furnace is hotter, why does it correspond to a lower density?
  5. In this paper, there is no in-depth analysis and explanation of the reasons for the non-uniformity of temperature, that is, the relationship between temperature and position, and the mechanism of the effect of temperature and growth atmosphere on the growth of single-walled carbon nanotubes. Please add.
  6. The sentence in lines 260 and 261 is incorrect. The lower thermal environment in the region of the highest tube density results in a wider size distribution of catalytic nanoparticles and corresponding tubes.
  7. At the end of section 3.4, please analyze how the difference in visible light transmittance will affect the application of "as a transparent conductive film".
  8. Line 334, how to interpret "an enhanced thermal environment in the chamber" ? Doesn't CVD center have a higher temperature?

Author Response

Taken into account all the following shortcomings mentioned, this paper needs to be revised.
1. Language expression. Please polish the language and check for grammatical errors to make the sentence more readable and logical. --> The manuscript is revised by native speaker as suggested by the reviewer 2.

2. In Figures 2 and 4, why didn't you add SEM images of the -12.5~-5 cm location? --> We thank for the good comment. The SWNTs were not grown at that position.

3. The temperature inhomogeneity at the horizontal position of the CVD furnace is not a favorable factor
for the growth of single-walled carbon nanotubes, especially for the batch growth of single-walled carbon nanotubes.
Can a reasonable method be proposed to optimize the temperature distribution? What is the innovation of this article? --> First, we thank for the nice comment and also fully agree with the Reviewer 2.
The temperature homogeneity is essential for the reliable work and that can be achieved by the stable heat source.
At this moment, the inlet and outlet parts are rapidly cooling down by air convection.
Thus, the temperature decrease at both edges is unavoidable.
From this context, some works recently adopted rapid heating by using such as Xenon Flash Lamp.
This approach is somewhat different from our present study. Here, we simply would like to clarify
the growth behavior of SWNTs with different sample position along the axial direction in a horizontal CVD chamber.

4. In lines 138 and 139, the expression of the sentence does not seem to be very accurate.
The middle of the CVD furnace is hotter, why does it correspond to a lower density? --> We thank for the poignant comment of the Reviewer 2.
We fully accept the comment and modified the vague original expression to evidently describe the situation.
The main factor to decide the growth yield of SWNT is not the simple temperature measured but the actual thermal environment,
which is closely related with the gas flow and partial pressure of carbon species at specific position.
In addition, the same growth trend can be seen from some reports of vertical growth of MWNTs,
which are added as references [Ref 13-17] in the revised version.

5. In this paper, there is no in-depth analysis and explanation of the reasons for the non-uniformity of temperature,
that is, the relationship between temperature and position, and the mechanism of the effect of temperature and growth
atmosphere on the growth of single-walled carbon nanotubes. Please add. --> We are sorry for unclear description at this part. The same answer to the 4th comment is available.
We modified the original manuscript to express clearly.

6. The sentence in lines 260 and 261 is incorrect. The lower thermal environment in the region of
the highest tube density results in a wider size distribution of catalytic nanoparticles and corresponding tubes. --> We thank for the comment. The higher thermal environment produces a wide size distribution of catalytic nanoparticles
and resultant SWNTs as well. The color showing RBM region in Fig. 6 is modified in the revised version.

7. At the end of section 3.4, please analyze how the difference in visible light transmittance will
affect the application of "as a transparent conductive film". --> Generally, high transmittance is better to apply as a transparent conductive film. In this study, high density of SWNTs
yields a relatively lower transmittance 96.8%. As a further work, we are investigating the trade-off effect between
transmittance and planar conductivity in order to fabricate the relevant devices.

8. Line 334, how to interpret "an enhanced thermal environment in the chamber" ? Doesn't CVD center have a higher temperature? --> We are sorry for unclear description at this part. The same answer to the 4th and 5th comments is available.
We modified the original manuscript to express clearly.

Reviewer 3 Report

This manuscript reports how the substrate position inside the horizontal furnace affects the SWCNT yield. The authors did a detailed investigation of such influence under various total gas flow rates and growth temperatures and found same trend: the highest yield of SWCNTs was achieved in the rear region of the reactor, not the central region. Although the manuscript is nicely organized and well presented, it lacks novelty. The findings are generally empirically well known in the synthesis field and there are already some articles reporting this phenomenon long time ago as well, but the authors did cite any. Here are some of the references: Carbon 48 (2010) 2106-2122, J. Phys. Chem. B 110 (2006) 8250-8257, ACS Nano 7 (2013) 3565-3580. Therefore, the reviewer does not recommend publication in Nanomaterials.

Author Response

We are very appreciate the reviewer’s opinion that recognize our detailed investigation and fruitful comments. We are well aware of the previous reports from J. Hart group but, honestly the first article (Carbon 48 (2010) 2106) is not familiar to us. The main difference of our study from others is that we grow not vertically aligned multi-walled carbon nanotubes (VACNT) but SWNTs. The growth feature of SWNTs are significantly influenced even by the minute change of the various experimental parameters, which is quite different from VACNT growth cases. Anyhow, the authors thank again for the comment and added suggested papers in ‘References’ section.

Reviewer 4 Report

  1. Ref [12] is not refered in the text of the manuscript
  2. Your statement about 'first systematic investigation'  is questionable (https://doi.org/10.1016/j.carbon.2011.08.003, https://doi.org/10.1016/S1001-0521(08)60197-7 for example)
  3. In chapter 3.2 you report confirmation of previous results by variation of gas composition, but you used H2:CH4 = 1:9 only with different flow rates. Fix it, please.
  4. Fix the title of the paper [1] in Reference section

Author Response

1. Ref [12] is not refered in the text of the manuscript
--> We thank for the kind comment and revised the original text as suggested.

2. Your statement about 'first systematic investigation' is questionable (https://doi.org/10.1016/j.carbon.2011.08.003, https://doi.org/10.1016/S1001-0521(08)60197-7 for example) --> We appreciate for helpful comment and revised the original manuscript.
In additioin, some papers including 2 articles suggested by the Reviewer 4 are
added as ’References’ in the revised version.

3. In chapter 3.2 you report confirmation of previous results by variation of gas composition, but you used H2:CH4 = 1:9 only with different flow rates. Fix it, please. --> We appreciate for kind comment and delete the expression.

4. Fix the title of the paper [1] in Reference section --> We are sorry for the mistake and fix the title of Ref [1].

Round 2

Reviewer 2 Report

This paper told us that the yield of single-walled carbon nanotubes in the horizontal CVD chamber varied with the position of the sample, and it was found that the yield of single-walled carbon nanotubes was the highest at the back of the horizontal reactor. The results obtained could be used to guide the synthesis of various functional nanomaterials in the horizontal CVD chamber.

According to the suggestion before, the authors have revised this article including the language such as grammar and format, interpunction, the authors also gave the reasonable expression about the questions.

The manuscript can be accepted for publication.

Reviewer 3 Report

Small revisions (only two sentences) were made in the introduction. Although the CNTs in this work are SWCNTs but not MWCNTs, the scientific mechanism behind is basically the same, regardless of the materials grown, which has been previously reported long time ago. Based on the reviewer’s knowledge, this work still lacks novelty and doesn’t add new scientific insight. Therefore, the reviewer does not recommend publication in Nanomaterials.